# What Can De Novo Protein Design Bring to the Treatment of Hematological Disorders?

**DOI:** 10.3390/biology12020166

**Published:** 2023-01-20

**Authors:** Hui Lu, Zhipeng Cheng, Yu Hu, Liang V. Tang

**Affiliations:** Department of Hematology, Union Hospital, Tongji Medical College, Huazhong University of Science and Technology, Wuhan 430022, China

**Keywords:** de novo protein design, protein therapeutics, hematological disorders

## Abstract

**Simple Summary:**

Because of the seriousness and complexity of hematological disorders, it is particularly critical to develop new methods for treating them. A protein engineering technique has been used to further enhance therapeutic effects and minimize side effects of protein-based therapeutics. However, the essence of the protein engineering technique is to modify and/or ameliorate natural existing proteins. In recent years, de novo proteins have been developed at a high speed, and their applications in the biomedical field are increasing, including in developing novel diagnostic and therapeutic drugs, novel vaccine techniques, and novel biological materials. At the same time, de novo proteins have also been applied to improve the efficacy of treatment methods in hematological disorders, such as designing the novel structures of chimeric antigen receptors, new inhibitors for treating chronic myeloid leukemia, and the novel type of interleukin-2. Therefore, the purpose of our review is to summarize the recent development of de novo protein design and its application in biomedicine, especially in exploring new treatment methods for hematological disorders.

**Abstract:**

Protein therapeutics have been widely used to treat hematological disorders. With the advent of de novo protein design, protein therapeutics are not limited to ameliorating natural proteins but also produce novel protein sequences, folds, and functions with shapes and functions customized to bind to the therapeutic targets. De novo protein techniques have been widely used biomedically to design novel diagnostic and therapeutic drugs, novel vaccines, and novel biological materials. In addition, de novo protein design has provided new options for treating hematological disorders. Scientists have designed protein switches called Colocalization-dependent Latching Orthogonal Cage–Key pRoteins (Co-LOCKR) that perform computations on the surface of cells. De novo designed molecules exhibit a better capacity than the currently available tyrosine kinase inhibitors in chronic myeloid leukemia therapy. De novo designed protein neoleukin-2/15 enhances chimeric antigen receptor T-cell activity. This new technique has great biomedical potential, especially in exploring new treatment methods for hematological disorders. This review discusses the development of de novo protein design and its biological applications, with emphasis on the treatment of hematological disorders.

## 1. Introduction

The number of functionally distinct proteins is much larger than that of genes, because procedures such as alternative splicing of genes and post-translational modification of proteins are needed during protein synthesis [1,2,3]. Moreover, proteins have diverse and significant roles in the human body to maintain normal physiological functions, such as catalyzing biochemical reactions, forming receptors and channels in membranes, and providing intracellular and extracellular scaffolding support [4].

Protein-based therapeutics have been widely used to treat hematological disorders. One strategy is to replace proteins that are deficient or abnormal, such as factor VIII, factor IX, and human albumin [5,6,7]. Another aim is to augment existing pathways, such as erythropoietin, tissue plasminogen activator, and interferon [8,9,10]. Another approach is to interfere with molecules or organisms, such as rituximab, used to treat CD20+ B-non-Hodgkin lymphoma, and basiliximab, used in graft-versus-host disease prophylaxis during hematopoietic stem cell transplantation [11,12]. Earlier protein therapeutics relied on the purification of natural proteins. Owing to the deficiency of natural proteins, recombinant protein products are used to treat hematological disorders [13]. Over the past decade, protein engineering has led the development of protein therapeutics to treat hematological disorders. Moreover, many well-equipped protein products, such as bispecific antibodies, antibody–drug conjugates, and armored chimeric antigen receptor-T/natural killer (CAR-T/NK) cells, have improved the efficiency and minimized side effects of protein therapeutics [14,15,16,17,18].

Undoubtedly, great achievements have been made in traditional engineered protein therapeutics for the treatment of hematological disorders. The impact of protein engineering was acknowledged by the 2018 Nobel Prize for Chemistry [19]. Protein engineering refers to the targeted modification of already existing proteins, for instance, to enhance their biophysical or biochemical properties. However, even when using novel techniques to engineer proteins, the essence of protein engineering is to modify or alter the structure and function of existing natural proteins. Many problems remain to be overcome. First, traditional protein engineering approaches cannot produce any protein with the required functions because there is limited room for further modification and improvement of natural protein sequences and structures. Second, excessive modifications will lead to the loss of function [20]. Third, natural proteins are usually unstable. Changing the protein sequence through traditional protein engineering can lead to unfolding or aggregation. [20]. Thus, the limitations of traditional protein therapeutics concerning modification, inactivation, and instability are obstacles that could be difficult to overcome.

In the past few decades, computational de novo protein design had a far-reaching impact on biotechnology [21]. Many research efforts have advanced de novo protein design from an outrageous concept to a routine method. De novo protein design refers to designing proteins from scratch, rather than modifying or ameliorating natural existing proteins, which could produce novel protein sequences, folds, and functions with shapes and functions customized [22]. De novo protein design is based on physical interaction principles to design protein skeletons and protein complexes. Novel proteins generated by computational algorithms exhibit better stability and unique functions, which could solve the aforementioned challenges met in traditional protein therapeutics [23]. The recent and rapid development of this cutting-edge field has provided new treatment options for hematological disorders. The purpose of this review is to introduce the development of de novo protein design and its biological applications, particularly in the field of hematological disorder treatment. 

## 2. Development of De Novo Protein Design

Proteins mediate the basic processes of life in diverse ways. This fact has been the focus of many biomedical studies. Designing a protein from scratch is laborious when its sequence and exact structure are initially unknown. There are 20^200^ possible sequences for a typical-length protein design, since each of the protein’s 200 residues can be one of 20 amino acids [22]. With more knowledge of the basic principles of protein folding, protein biochemistry, and biophysics, it should be possible to design customized proteins from scratch. This would provide basic knowledge concerning how proteins work and could solve many important societal challenges. The physical rule of de novo protein design involves folding proteins in their lowest free energy state. The driving force of protein folding is embedded hydrophobic residues in the core of the protein, away from the solvent. The side chains in the core must be tightly packed with no energetically unfavorable atomic overlap to minimize the size of the cavity occupied by the protein in water and to maximize van der Waals forces. In addition, polar groups that interact with the solvent in the unfolded state buried during protein folding must form hydrogen bonds within the protein for compensation. If this does not occur, the huge energy cost of stripping water will not be conducive to folding [24]. 

As described by DeGrado, de novo design has evolved in three stages: manual protein design using a physical model, computational protein design guided by fundamental physicochemical principles, and fragment-based and bioinformatically informed computational protein design [25]. The definitions broadly overlapped with those described by Woolfson, namely, minimal protein design, rational protein design, and computational protein design [26]. The time-scale evolution of de novo protein design was summarized in detail by DeGrado and Woolfson [25,26].

Although de novo protein design has developed since the manual protein design stage, the goal of designing proteins with predetermined structures and functions has not yet been achieved. With continuous improvements and pivotal advances, the first protein with a predetermined structure of helical bundles was successfully designed from scratch [27]. Various helical bundles with specific structures were subsequently designed, such as four-helix bundles and coiled coils [28,29]. With the continuous development of de novo protein design with specific structures, scientists have begun to shift their attention to a new and as-yet elusive goal of designing proteins with predetermined functions rather than structures. The successful design of functional proteins from scratch provides an opportunity for their biomedical applications in the biomedical field. In the era of the computational protein design stage, the successful design of the TOP7 artificial 93-residue protein advanced de novo protein design to the use of backbone fragment libraries derived from protein data banks to build the backbones of de novo proteins [30].

Two steps are required for de novo protein design: First, the tertiary structure of the protein main chain is generated. Second, the amino acid identity and side chain conformation of all residue positions are determined. The current computational protein design stage involves the use of computers to find protein sequences folded into low-energy and thermodynamically favorable structures during protein design. Rosetta, one of the leading software suites for protein modeling and design, contains the most commonly used energy functions in protein design [31]. REF2015 is the current default Rosetta energy function, which has continuously improved over the past decades [32]. Recently, SCUBA, a statistical model that contains a backbone-centered energy function of neural networks, was developed for protein design [33]. Currently, there are three outstanding fragment-based methods enabling the design of the backbones of nonsymmetric structures: the blueprint builder, the topology builder, and structural extension [34,35,36].

## 3. Biomedical Applications of De Novo Protein Design 

With the continuous development of de novo protein design, significant progress has been made in designing proteins with specialized functions. Sufficient progress has been made in designing proteins with functions such as binding, catalysis, and crossing through membranes [37,38,39,40]. The advent of computational protein design has permitted access to a much larger protein sequence space than that sampled by evolution in nature [41,42]. Refinements in biophysics are enabling the creation of an increasing number of proteins with ideal structures, unique biochemical properties, and enhanced or even novel functions [43,44,45]. The de novo protein design technique has been widely applied in various fields. Here, we summarize the biomedical applications of de novo protein design (Table 1).

### 3.1. Novel Diagnostic and Therapeutic Drugs

Computational approaches have recently been applied in the design of new proteins and peptides for diagnostic and therapeutic applications in many diseases. Cysteine-rich protein 1 (CRIP1), an early biomarker for breast and other cancers, is difficult to detect using conventional antibody methods. Hao and colleagues developed a protocol generating high-affinity peptides (A1M) to purify cytosolic proteins to cross membranes and serve as imaging ligands by computational protein design techniques. The A1M peptide increased approximately a 10–28-fold affinity for CRIP1, which may help the early diagnosis of breast cancer [48]. Taghrid et al. have successfully designed a bioactive peptide analog with cytotoxic effects by using the Resonant Recognition Model. This peptide analog to a viral protein functions only on tumor cells and is suitable to be developed as a potential cancer therapeutic [67]. In addition to cancer, de novo protein design has also been applied to the treatment of virus-infected diseases. Floudas et al. designed human immunodeficiency virus (HIV)-1 entry inhibitors. The 12 amino acid–long peptide inhibitors target the hydrophobic core of gp41 and improve the inhibition effect against HIV-1 by 3- to 15-fold over the native sequence [53]. Pantazes et al. proposed a general computational method called Optimal Complementarity Determining Regions (OptCDR), which was used to design novel antibodies from scratch to bind selected antigens with high affinity and specificity. The authors applied OptCDR to design antibodies targeting a peptide from the capsid of the hepatitis C virus and obtained good results [54]. Rosetta has been used to design hemagglutinin inhibitors from the 1918 hemagglutinin 1 neuraminidase 1 (H1N1) pandemic virus, which also inhibits multiple other subtypes [56]. Virus entry inhibitors targeting the envelope protein of the dengue virus have also been identified [56]. Techniques have also been applied in the design of novel therapeutics for Alzheimer’s disease (AD). Using the Rosetta suite of tools, Sievers et al. designed and characterized an all-D amino acid inhibitor of fibrillation of the tau protein found in AD based on templates of atomic structures of segments of amyloid fibers [69]. Rajadas and colleagues designed a non-canonical and D-amino acid–containing peptide that organizes Aβ42 into stable oligomers, which could aid in the development of structural models for toxic oligomer formation and facilitate the development of treatment methods for AD [70]. 

### 3.2. Novel Vaccines

Based on the robustness and versatility of computationally designed proteins and the urgent need for vaccines to prevent virus-related disease, designing safe, effective, and scalable vaccines is very important. The coronavirus disease 2019 (COVID-19) pandemic is ongoing. Developing a vaccine against severe acute respiratory syndrome coronavirus 2 (SARS COV-2) is urgently needed to alleviate the public health, societal and economic impacts of the virus [89]. De novo protein design has an important role in vaccine development. Computationally designed protein nanoparticles have become promising platforms for multivalent antigen presentation. Walls et al. reported designed protein nanoparticle vaccines for SARS COV-2 that have proved to elicit potent and protective antibody responses in mice. Vaccine-elicited antibodies recognize multiple distinct receptor-binding domain (RBD) epitopes, making it difficult for the pathogen to escape from the immune system [72,73]. Vaccine candidates based on designed protein nanoparticles have also improved the potency and breadth of antibody responses against other antigens in preclinical studies, such as perfusion respiratory syncytial virus, HIV-1 envelope, influenza hemagglutinin, and *Plasmodium falciparum* cysteine-rich protective antigen [75,77,78]. De novo protein design has also been used to design cancer-targeting peptides for vaccines. Sundaram et al. successfully synthesized peptides derived from human T-cell leukemia virus type 1 (HTLV-1), a virus that may lead to adult T-cell leukemia/lymphoma, eliciting antibodies with neutralizing potential. A novel template design was used to construct two peptides, WCCR2T and CCR2T. Each peptide assembles into a triple helical coiled-coil conformation mimicking the gp21 crystal structure of HTLV-1 [79]. A lead trimeric peptide has been designed and identified as a cancer-targeting immunostimulatory peptide ligand to target and activate NK cell immunotoxicity directly toward tumors [80]. 

### 3.3. Novel Biological Materials

Protein-based materials have the potential to overcome many technical challenges. Different types of materials, such as lipids, dendrimers, carbon nanotubes, and metals, have been used as molecular carriers for the construction of nanoparticles [90]. Proteins are ideal biomaterials because of their natural structural roles, cost-effective biological features, functional modifications, and full biocompatibility [91]. Advances in de novo protein design have promoted the development of novel biomaterials. First, de novo designed protein building blocks have been used as functional nanoparticles and biologically safe vehicles that are urgently needed for imaging, drug delivery, and gene therapy [81]. These protein-only nanoparticles are based on the combined use of a cationic peptide and polyhistidine. The structural robustness and strong potential of end-terminal cationic tags have made them stable nano architectonic tools for medical applications [66]. Second, natural proteins have several complex and delicate structures. Among these proteins, cage-like multimeric proteins provide spatial control for biological processes and compartmentalize potentially toxic or unstable compounds to avoid contact with the environment [92]. Protein-based nanocages are of great interest because of their potential applicability as drug delivery carriers, perfect and complex symmetry, and ideal physical properties. Many natural structure-based protein nanocages, such as ferritin, small heat shock proteins, and vault proteins, have been discovered and used for drug delivery [93,94,95,96,97]. Promising research by many groups has laid the foundation for the application of designed protein-based nanocages. They have potential value for vaccine delivery and have shown potential as a substitute for gene and small molecule delivery [84]. Many teams have been committed to discovering de novo protein-based nanocages, which could enable the construction of nanocages with more complex structures, and continuous improvement of their functions [85,86]. Third, natural protein switches have been used to develop novel biosensors and reporters for clinical applications [98]. Biosensors have been used to detect a variety of substances, such as blood glucose, lactic acid, uric acid, urea, transaminase, etc., in the clinic. Similarly, de novo protein-based biosensors have been created by reversing the information flow through de novo designed protein switches, in which the binding of peptide bonds triggers biological outputs of interest [99]. When binding to the target of interest, it drives the analyte switching from a closed to an open state. Recently, de novo designed biosensors have been successfully created to sensitively detect the anti-apoptosis protein Bcl-2, the IgG1 Fc domain, the Her2 receptor, botulinum neurotoxin B, cardiac troponin I, and RBD of SARS COV-2 [59,87]. Fourth, complex protein nanomachines in nature have evolved to process energy and information by coupling biochemical free energy with mechanical work. Protein rotary machines, such as the F1 motor of adenosine triphosphatase or bacterial flagella, are the most studied and complex class of protein nanomachines [100,101,102]. Recent studies have shown that computational protein design methods enable bottom-up exploration of de novo construction of the protein machinery [88]. These designed mechanical systems provide opportunities for genetically encodable nanomachines [90]. To sum up, de novo protein design provides a method for engineering various nanodevices for medicine, material sciences, and industrial bioprocesses. 

## 4. De Novo Protein Design in Treating Hematological Disorders

Although protein therapeutics have been widely used in treating hematological disorders, the aforementioned limitations and obstacles still restrict their further applications. With the advent of computational de novo protein design, protein therapeutics for treating hematological disorders seeks to produce novel protein sequences, folds, and functions with shapes and functions customized to bind therapeutic targets. Here we discuss some new de novo protein design products that hold great promise for treating hematological disorders in the future.

### 4.1. Colocalization-Dependent Protein Switches-Latching Orthogonal Cage-Key pRotein (Co-LOCKR) for CAR-T Cell Therapy

CAR-T cell therapy, as a novel and promising cellular therapy, was first applied in treating hematological malignancies such as acute lymphocytic leukemia, chronic lymphocytic leukemia, and multiple myeloma [103,104]. CAR-T cell therapy has gained much attention and has made remarkable achievements in treating hematological disorders [105,106]. Anti-CD19 and anti-BCMA CAR-T cell therapies have been approved by the Food and Drug Administration (FDA) [107]. However, owing to the lack of a single antigen marker to distinguish tumor cells from healthy tissues, the off-target effect is still a challenge in conventional CAR-T cell therapy [108]. For a long time, researchers’ efforts have addressed the structural engineering of CAR to overcome the off-target side effects and enhance tumor specificity. 

The logic gating control is used for designing the CAR structure [109]. As mentioned above, considerable progress has been made in the de novo protein design of stable protein structures based on the principle of protein folding to its lowest free energy state. Baker et al. have explored the de novo design of switchable protein systems by regulating intermolecular and intramolecular interactions [100]. 

Baker et al. designed Co-LOCKR, which is based on a de novo designed precursor switch called LOCKR (Figure 1). LOCKR consists of a static, five-helix “Cage” with a single interface that can interact with a terminal “Latch” helix or a specific peptide partner “Key”. The Latch, which is always a functional motif for binding, degradation, or nuclear export, works when the Key frees the Latch from the Cage. The Key can competitively bind to the Cage and release the Latch, thereby rendering a previously buried signal motif accessible to putative binding partners (Bcl2–Bim binding) [100]. The Bim–Bcl2 interaction, the core of apoptosis, was selected as the model system for installing function into LOCKR. Bim is caged so that its binding to Bcl2 occurs only in the presence of the Key [110]. Co-LOCKR works in a colocalization-dependent manner. The designer ankyrin repeat proteins (DARPin) are fused to the Key and Cage for binding antigen targets. Therefore, Co-LOCKR could perform “AND,” “OR,” and “NOT” Boolean logic operations [111]. These de novo proteins perform computations on the surface of cells and are activated once upon detection of specific cell surface antigens, which provides much precision in CAR-T cell therapy. Computing logic operations “1 AND 2” on the surface of cells could increase targeting selectivity; “1 AND (2 OR 3)” could help target heterogeneous tissue; and “1 and 2 not 3” could help to avoid targeting healthy tissues [111]. This advance could help overcome adverse effects, such as B-cell aplasia in anti-CD19/CD20 CAR-T cell treatment [108,112]. The newly designed CAR-T cell approaches against K562 and Raji cells have demonstrated the potential of Co-LOCKR to mediate targeting specificity in vitro. However, some questions still need to be addressed for Co-LOCKR to become a clinically transformable therapy. In vivo studies are needed to evaluate the safety and improve the pharmacokinetics of Co-LOCKR components. The immunogenicity of the designed protein is also a potential problem.

### 4.2. De Novo Designed Molecules

Chronic myeloid leukemia (CML) is a clonal myeloproliferative disorder that affects blood and bone marrow. CML was the first cancer that was directly related to genetic abnormalities. Since the first kinase inhibitor, imatinib, received FDA approval in 2001, an increasing number of kinase inhibitors have been investigated for treating CML [113]. However, challenges, such as a high failure rate during development, side effects, and drug-resistance problems, still exist during the discovery of kinase inhibitors [114]. The novel method of de novo protein design has been applied to solve these problems by obtaining new tyrosine kinase inhibitors. Based on the calculated GScore, de novo designed molecules exhibit a better capacity than the currently available tyrosine kinase inhibitors. These molecules have the potential to become drugs capable of inhibiting all mutations in CML, particularly the BCR-ABL-T315I mutant [115]. The novel synthetic drug created using fragment-based drug design could be a good alternative for treating CML.

### 4.3. Neoleukin-2/15 (Neo-2/15)

Interleukin-2 (IL-2) is a potentially valuable compound for cancer treatment. Accordingly, numerous efforts, such as mutation and/or chemical modification, have been made to further improve the therapeutic properties of IL-2 [116,117,118,119,120]. Recombinant IL-2 has been approved for the treatment of melanoma and renal cell carcinoma [121]. 

The IL-2 soluble glycoprotein is produced by lymphocytes and other cells following activation by antigens or mitogens. To achieve biological effects, IL-2 needs to be combined with IL-2 receptor (IL-2R) expressed on target cells, such as effector T cells and NK cells. IL-2R consists of three subunits: IL-2Rα (CD25), IL-2Rβ, and IL-2Rγ. The β and γ chains participate in intracellular signal transduction, whereas the α chain does not. IL-2Rα alone has a low affinity for IL-2 and cannot transduce signals. According to their affinity for IL-2, IL-2R is divided into low- (α chain), medium- (β or βγ), and high-affinity (αβγ or αβ) receptors. The combination of IL-2 and IL-2Rβγ initiates the subsequent signal cascade reaction, which can stimulate immune cells with anti-tumor functions. However, IL-2Rαβγ (the affinity for IL-2 is higher than that for IL-2Rβγ) is expressed on the surface of some off-target cells, such as endothelial cells and immunosuppressive regulatory T cells. These off-target CD25+ cells may respond to high-dose IL-2 therapy with much higher potency than the intended target cells. Consequently, serious side effects occur [104]. Owing to the off-target side effects of IL-2 caused by IL-2Rα (CD25), many efforts have sought to eliminate or reduce its natural binding preference for CD25 and preserve its binding ability to IL-2RβƔc [122,123]. 

Based on the current Rosetta design methodology, Baker et al. built fully de novo proteins, neoleukin-2/15 (Neo-2/15), the first de novo protein immunotherapeutic. Neo-12/15 designed by this method has the ability to bind to IL-2Rβγ but has no binding site for IL-2Rα or IL-15Rα (Figure 2). 

Neo-12/15 is an experimentally optimized mimic of IL-2/15, with superior therapeutic activity compared with IL-2 in murine models, with reduced toxicity and undetectable immunogenicity [124]. Neo-2/15 is a highly thermostable and tunable protein that demonstrates potent signaling in both human and mouse cells expressing IL-2Rβγ receptors. These properties allow the routine use of Neo-2/15 for indications for which IL-2 is not broadly used, such as to enhance the function of CAR-T cells [124]. The newly designed protein can also be used to treat hematological disorders through CAR-T cell therapy.

## 5. Discussions, Success and Challenges

With the rapid development of de novo protein design, creating de novo proteins with extreme stability, diverse shapes, and customized functions has gradually become a readily available technique. Compared with the engineered natural proteins, de novo proteins could be smaller and more stable, easy to produce, and convenient to adjust. Over the past two decades, computational protein engineering has made great progress biomedically, including the design of novel diagnostic and therapeutic drugs, novel vaccines, and the synthesis of novel biological materials. The de novo protein technique also provides new options for the treatment of hematological disorders. The highly versatile Co-LOCKR approach provides an important asset for “universal” CAR-T therapies, which could function in minimizing the off-target effects [110]. De novo inhibitors also exhibit better inhibitor capacity than tyrosine kinase inhibitors for treating CML [116]. Neo-2/15 can enhance the functions of CAR-T cells with reduced toxicity and undetectable immunogenicity [124].

Although progress has been made, challenges remain. Improving the computational efficiency of de novo design algorithms is essential because of the complexity of the proteins. In addition, while there are a growing number of de novo proteins that bind targets with agonistic or antagonistic properties, the routine and accurate design of specific and tight binding still rely on the availability of high-quality structural information of the targets. Therefore, further advancements in protein structure/dynamics prediction and determination are necessary to enhance the capabilities of de novo protein design [125,126,127]. Moreover, the problem of immunogenicity is still a potent challenge for de novo proteins when administrated to humans. The recombinant protein could elicit the immunogenic response, so we reasonably speculate that de novo proteins may lead to a strong immunogenic response for their low sequence homology with natural proteins [128,129].

In general, there is no doubt that the de novo protein design has achieved great success in the biological field, including in treating hematological disorders. However, due to the challenges we summarized above, efforts are still needed to overcome these difficulties. In the near future, Co-LOCKR may also have the possibility to apply in CAR-NK cell therapy, another emerging cell-based therapy method. Furthermore, in addition to synthesizing cytokine mimics, bioactive molecules with known or accurately predictable structures will be synthesized, allowing for enhanced therapeutic properties and reduced side effects. De novo protein design provides an innovative method to overcome the inherent limitations of natural protein building blocks, opening the door to a new class of therapeutic methods in hematological disorders. In the next few years, emerging progress will mark technological progress similar to the transition from the Stone Age to the Iron Age.

## 6. Conclusions

The development of de novo protein design has indeed brought us great surprises. The evolution of de novo protein design has experienced three stages: minimal protein design, rational protein design, and computational protein design. Proteins with predetermined structures and specialized functions could be designed and synthesized currently. This review mainly discusses the development of de novo protein design and its biological applications, with emphasis on the treatment of hematological disorders. In the biomedical field, the de novo protein design technique has been applied in designing novel diagnostic and therapeutic drugs, novel vaccines, and novel biological materials. Moreover, Co-LOCKR, de novo designed molecules and Neo-2/15 also function in protein-based therapeutics for hematological disorders. We believe that continuous progress will be made in de novo protein design and that the challenges faced today can be overcome in the near future.

## Figures and Tables

**Figure 1 biology-12-00166-f001:**
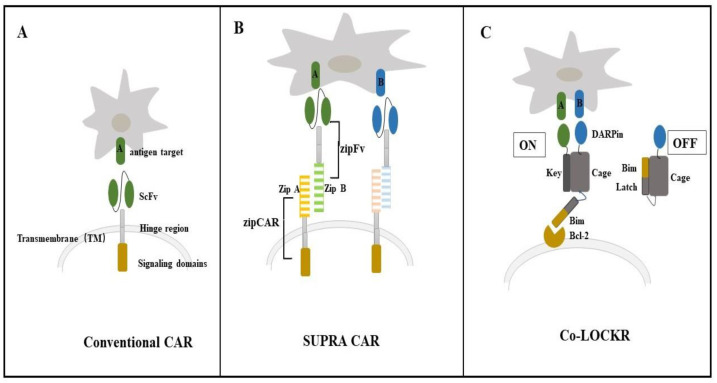
Different chimeric antigen receptor (CAR) strategies performing Boolean logic operations to avoid the off-target side effect. (**A**) Conventional CAR structure. (**B**) The split, universal, and programmable (SUPRA) CAR is a receptor system consisting of a universal receptor (zipCAR) expressed on T cells and a tumor-targeting scFv adaptor (zipFv). The design of SUPRA CAR is based on the naturally existing leucine zipper structure. (**C**) Co-LOCKR is based on de novo protein design.

**Figure 2 biology-12-00166-f002:**
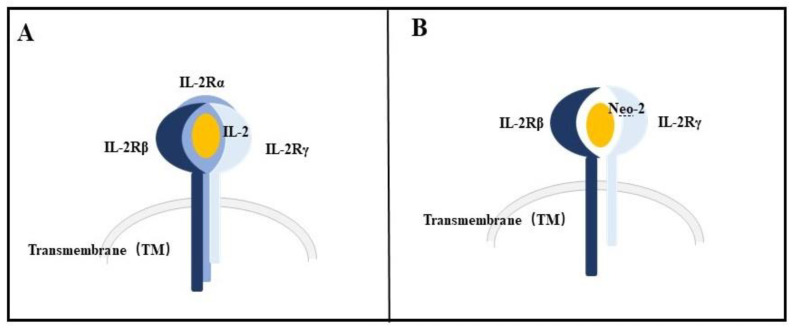
The comparison between the binding sites of IL-2 (**A**) and Neo-2 (**B**).

**Table 1 biology-12-00166-t001:** The applications of de novo protein design in the biomedical field.

Type	Application	Summary	Reference
Novel diagnostic drugs	infectious diseases	De novo designed protein could sensitively detect diverse targets, such as salmonella typhi TolC protein, anti-hepatitis B antibodies, botulinum neurotoxin B, and SARS COV-2.	[46,47]
	neoplastic diseases	A high-affinity peptide (A1M) has been designed by computational protein design techniques, which may help the early diagnosis of breast cancer.	[48]
Novel therapeutic drugs	infectious diseases	Bacterial infectious diseases: various de novo antimicrobial peptides have been designed for the treatment of bacteria-induced infections. Viral infectious diseases: various de novo designed inhibitors have been used for the treatment of viral infectious diseases, such as HIV, hepatitis C, H1N1, dengue fever, and COVID-19.	[49,50,51,52,53,54,55,56,57,58,59,60,61]
	neoplastic diseases	Various de novo designed inhibitors, such as histone methyltransferase inhibitors, ROCK inhibitors, SHP2 inhibitors, BRAF kinase inhibitors, and Cdc25B inhibitors, could be considered as anticancer agents for further investigation.	[62,63,64,65,66,67]
	other diseases	De novo peptides, such as all-D-amino acid inhibitors and β-amyloid peptide inhibitors have been designed for the treatment of AD. De novo protein tyrosine phosphatase 1B has been designed for the treatment of type-2 diabetes mellitus.	[68,69,70,71]
Novel vaccines	protein nanoparticle vaccines	Designed protein nanoparticles have been used as vaccines for SARS COV-2, perfusion respiratory syncytial virus, HIV-1 envelope, influenza hemagglutinin, and Plasmodium falciparum cysteine–rich protective antigen.	[72,73,74,75,76,77,78]
	cancer-targeting peptide vaccines	Designed peptides could be used as vaccines for adult T-cell leukemia/lymphoma. A designed trimeric peptide could target and activate NK cell immunotoxicity directly toward tumors.	[79,80]
Novel biological materials	designed protein-based nanoparticles and vehicles	Designed protein-based nanoparticles and vehicles could be used for imaging, drug delivery, and gene therapy.	[81,82,83]
	designed protein-based nanocages	Designed protein-based nanocages could be used for vaccine, gene, and small molecule delivery.	[84,85]
	designed protein-based biosensors	Designed biosensors could be used to detect the anti-apoptosis protein Bcl-2, the IgG1 Fc domain, the Her2 receptor, botulinum neurotoxin B, cardiac Troponin I, and RBD of SARS-COV-2.	[59,86,87]
	designed protein-based machinery	Designed mechanical systems provide opportunities for genetically encodable nanomachines.	[88]

## Data Availability

No data have been generated for this article; all the data are cited from published literature.

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
