# Peer review of "What Can De Novo Protein Design Bring to the Treatment of Hematological Disorders?"

_biology, 2023, doi:10.3390/biology12020166_

Round 1

Reviewer 1 Report

This review would, ideally, focus on an interesting topic about AML. Nevertheless, it's almost impossible to understand the meaning of the review  because of two main reasons:

- English is almost incomprehensible. Example: Chronic myeloid leukemia (CML) is related to hematopoietic stem cells, which are mainly characterized by bone marrow hyperplasia. 

- Each section contains too many information, I don't get the point. It's hard to find a "take home message" from this work.

I suggest to extensively revise the paper, then re-submit. 

Reviewer 2 Report

In this review, Lu et al. summarized applications of de novo protein designs in the biomedical field and highlighted its impact in treating hematological disorders. De novo protein design is for creating novel proteins with endowed biological functions. It is of great significance in many scientific fields to enable new functions or compensate for natural defects. The review could provide an overview of the achievements of de novo-designed proteins in biomedical applications.

The major issue of the manuscript is the lack of clarity in some logical organizations. Many examples were mentioned, however, de novo protein designs were just part of their story. The authors should describe clearly which part was designed by de novo protein designs. For example, CAR-T therapy was the main application of protein therapeutics described in hematological disorders. The design of conventional CAR is not belonging to de novo protein designs. De novo-designed protein is just the ‘LOCKER’ module. This should be highlighted to emphasize the feature of de novo protein designs. Besides, we all know that the de novo protein designs remain challenging as the biological functions of natural proteins are mediated by irregular structures. The authors should involve an individual part to review challenges in the manuscript to let readers in different fields understand it.

Specific comments:

1. The abstract could be briefer.

2. In the introduction part, the definition of de novo protein designs should be added. This is important for helping readers to know the emphasis of the manuscript and the relationship between de novo protein designs and protein engineering.

3. In line 93, the calculation is incorrect. For a protein with n residues, the possible sequence should be 20n.

4. An individual part should be added to review the general success and challenges of de novo protein designs.

5. Is there a de novo protein design strategy in the example of ‘SUPRA CAR’? If not, the example should be removed.

6. The ‘LOCKER’ description (282-300) is unclear.

Reviewer 3 Report

The authors have well explained protein designs which are the next generation to therapeutics. De-novo designs of protein can help in targeting different mis-regulated proteins and can be widely used in hematological disorders. I would like the authors to add a table describing the development of denovo-protein design in biomedical field.  Further a mechanistic figure in the evolution of the denovo protein design with time scale can add to the review work.

The title is not interesting. I highly recommend the title to be changed.

Round 2

Reviewer 1 Report

The authors provided an updated paper that is now less focused on leukemia, and provides an overview on protein products and their potential applications in hematological disorders in general. The whole paper is better organized and interesting overall. There are still some minor issues with English all around; I suggest to revise it. 

Reviewer 2 Report

The authors have addressed my concerns in the revised manuscript. Some more comments to further improve the manuscript are as follows:

1. The introduction of the limitations of protein engineering (lines 65-73) should be revised. Since they are not common problems in protein engineering. Besides, de novo protein designs also have the same problems.

2. There are repetitive sentences in the text and the added table. For example, the description of biological materials in lines 214-216 was exactly repeated in the table.

3. The descriptions in the table should be briefer and concise. Several citations in the table were reported about 10 years ago. They should not be described as ‘recent biomedical applications’ (line 145). If there are more recent studies, please cite them.
